# Ultrasound-Guided Botulinum Toxin-A Injections into the Masseter Muscle for Both Medical and Aesthetic Purposes

**DOI:** 10.3390/toxins16100413

**Published:** 2024-09-24

**Authors:** Marius Nicolae Popescu, Cristina Beiu, Carmen Andrada Iliescu, Andreea Racoviță, Mihai Berteanu, Mădălina Gabriela Iliescu, Ana Maria Alexandra Stănescu, Diana Sabina Radaschin, Liliana Gabriela Popa

**Affiliations:** 1Department of Physical and Rehabilitation Medicine, Elias Emergency University Hospital, Carol Davila University of Medicine and Pharmacy, 020021 Bucharest, Romania; marius.popescu@umfcd.ro (M.N.P.); mberteanu@gmail.com (M.B.); 2Department of Oncologic Dermatology, Elias Emergency University Hospital, Carol Davila University of Medicine and Pharmacy, 020021 Bucharest, Romania; 3Clinic of Dermatology, Elias Emergency University Hospital, 011461 Bucharest, Romania; andreea.stefania.racovita@gmail.com; 4Department of Medical Rehabilitation, Ovidius University, 900527 Constanța, Romania; iliescumadalina@gmail.com; 5Department of Family Medicine, Carol Davila University of Medicine and Pharmacy, 020021 Bucharest, Romania; alexandrazotta@yahoo.com; 6Department of Clinical Medical, Faculty of Medicine and Pharmacy, Dunărea de Jos University, 800385 Galați, Romania; dianaradaschin@yahoo.com

**Keywords:** ultrasound-guided injection, botulinum toxin type A (BoNT-A), masseter muscle hypertrophy

## Abstract

With the increasing use of Botulinum toxin type A (BoNT-A) injections in the masseter muscles for both medical and aesthetic purposes, there is a constant need to continually enhance the efficacy of these treatments and reduce the risk of potential adverse events. This review provides an in-depth analysis of the masseter muscle’s anatomical structure and essential landmarks and emphasizes the advantages of ultrasound (US) guidance in improving the precision of BoNT-A injections compared to conventional blind methods. The review is supplemented with comprehensive figures, including graphics, clinical images, and ultrasound visuals, to support the discussion. Potential complications such as paradoxical bulging, inadvertent injections into the risorius muscle or parotid gland, facial paralysis, and the risk of bone resorption are examined. Future research should aim at refining injection techniques and assessing the long-term effects of repeated treatments to ensure optimal patient care and safety.

## 1. Introduction

The integration of botulinum toxin type A (BoNT-A) into therapeutic protocols for managing various masseter muscle conditions, such as masseter hypertrophy, bruxism, and specific temporomandibular disorders (TMDs), represents a notable development in contemporary medical treatments [1,2]. Conditions involving the masseter muscle are known to impair daily functionalities, such as chewing, speaking, and swallowing, leading to pain and discomfort, stiffness, and limited movement of the jaw [3]. Noteworthy research, including randomized controlled trials, has provided evidence supporting the effective use of BoNT-A, demonstrating marked improvements in muscle hyperactivity, pain reduction, and an increase in the range of motion in conditions related to the masseter muscle [4,5]. Furthermore, BoNT-A injections in the masseter have also extended to managing oromandibular dystonia [6,7], myofascial pain syndrome (MPS) [8,9], and even tension-type headaches (TTH) [10,11].

Concurrently, the aesthetic enhancement of the facial contour, particularly achieving a slimmer and smoother jawline, has emerged as a highly desirable trait, leading to an increased application of BoNT-A for cosmetic purposes [12,13,14]. This dual utility of BoNT-A underscores a growing interest in optimizing the safety and efficacy of its use.

BoNT-A works by blocking the release of acetylcholine, a neurotransmitter crucial for muscle contraction at the neuromuscular junction. This action causes temporary muscle paralysis, leading to muscle atrophy and a reduction in muscle size. BoNT-A is often used for therapeutic and cosmetic purposes due to its minimally invasive nature, high efficacy, and low complication rates [15,16]. Nevertheless, despite its generally favorable safety profile, BoNT-A injections carry inherent risks, primarily due to the complex and variable anatomy of the facial region. These risks stem from the intricate structures and anatomical differences at the injection sites among individuals [17].

The utilization of ultrasound (US) guidance in administering BoNT-A injections emerges as an essential technique in mitigating risks [18,19]. This method allows for real-time visualization of the injection site, ensuring accurate delivery of BoNT-A to the intended muscle group while avoiding surrounding structures, thus minimizing adverse effects.

Therefore, the aim of this comprehensive review is to examine the role of the masseter muscle, effective injection techniques, the use of US guidance, potential adverse effects, and future directions for repeated BoNT-A injections for both medical and aesthetic applications.

## 2. Masseter Muscle—General Considerations

Masseter muscle, along with the temporalis, medial, and lateral pterygoid, represents a group of interconnected masticatory muscles that contribute to grinding movements. Out of these four, the masseter is the largest one, with its volume being determined by lifestyle habits, biting force, external stressors, or motor behaviors [14]. Any of these can lead to masseter hypertrophy, which is defined as the unilateral or bilateral enlargement of the muscle over the angle of the mandibula. It can be asymptomatic or contribute not only to bruxism, functional impairment, or discomfort but also to aesthetic disharmony [20]. Traditional approaches, such as surgical muscle resection, liposuction of the subcutaneous local fat, or mandible reduction, are no longer the standard, allowing for more conservative alternatives [14].

BoNT-A injection is a non-invasive alternative procedure that has emerged as a viable therapeutic solution for a range of medical conditions associated with the masseter muscle, including masseter hypertrophy, bruxism, TMDs, oromandibular dystonia, MPS, or TTH [21,22,23].

Bruxism involves involuntary and excessive grinding of teeth and clenching of the jaw, often leading to jaw pain, headache, and dental damage. BoNT-A injections in the masseter reduce muscle contractions, thereby decreasing the force exerted during grinding and clenching episodes, with several studies noting improvement in both symptoms and dental outcomes [21].

Furthermore, BoNT-A’s reduction of masseter hypertonicity can significantly alleviate joint stress and improve jaw function, with randomized controlled trials supporting its use for treating TMDs that can arise as a consequence of bruxism or independently [22,23].

It was also shown that BoNT-A injections are useful in treating oromandibular dystonia, a movement disorder presenting as involuntary muscle contractions affecting the jaw, lips, and tongue. BoNT-A injections provide symptomatic relief by reducing muscle spasms and associated pain and functional impairment [7,24]. Moreover, BoNT-A treatment targeting the masseter has demonstrated effectiveness in treating other facial spasms, such as hemifacial spasms and blepharospasm, with high success rates noted in controlled studies [25,26]. Its utility in managing Meige syndrome, characterized by oromandibular dystonia accompanied by blepharospasm, underscores BoNT-A’s wide applicability in treating an array of conditions involving the masseter muscle [27].

Myofascial pain syndrome (MPS) involving the masseter muscle is a chronic pain condition characterized by the presence of trigger points in the masseter muscle, leading to localized pain and discomfort, often with referred pain in the jaw, head, or neck [28]. The effectiveness of botulinum toxin in alleviating MPS symptoms is attributed to its ability to reduce muscle activity and thus relieve the tension and trigger points that cause pain [29]. Chronic facial pain associated with masticatory hyperactivity can also be managed with BoNT-A. In a randomized placebo-controlled trial involving 90 patients, those treated with BoNT-A showed a significant decrease in pain, with an average reduction of 3.2 points on a visual analog pain scale—a statistically significant improvement compared to the placebo group (*p* < 0.01). These findings indicated that BoNT-A injections are an effective treatment option for patients unresponsive to conventional therapies, with approximately 90% experiencing symptom relief [30].

Tension-type headaches (TTH) are often linked to muscle tension in the head and neck, particularly in the masseter muscle. A systematic review on the use of BoNT-A for TTH that included a total of 899 patients across 11 studies concluded that BoNT-A injections can significantly reduce the intensity and frequency of TTH, underscoring that it offers a promising alternative, especially for patients not responding to conventional treatments like analgesics or muscle relaxants [31].

In addition to their well-documented medical benefits, BoNT-A masseter injections have also proven highly effective in aesthetic applications. Studies have consistently demonstrated that BoNT-A injections effectively reduce masseter muscle volume and thickness, significantly enhancing facial symmetry and refining facial contours without the need for invasive procedures. This non-surgical approach allows for the achievement of a more balanced facial appearance and the attainment of desired aesthetic outcomes [20,32,33,34]. Additionally, research underscores the efficacy of BoNT-A injections in reshaping the jawline [35,36]. These treatments not only reduce the size of the masseter muscles but also elevate the jawline, enhancing symmetry and contributing to a more defined and sculpted appearance. This dual effect of improving jawline definition and facial symmetry underscores the growing popularity of BoNT-A injections in aesthetic medicine.

However, it is essential to consider the potential risks and side effects associated with these injections. When the toxin is injected into the masseter, it not only induces temporary paralysis of the targeted muscle fibers but may also cause atrophy in muscles adjacent to the injection site due to broad diffusion [37]. Moreover, imprecise placement of the neuromodulator or extensive infiltration can lead to undesired side effects, such as paradoxical bulging, asymmetric smile, xerostomia, and neurapraxia [37]. Therefore, an improved comprehension of the anatomical landmarks of the masseter muscle and surrounding area is mandatory for the safe and effective injection of BoNT-A [38]. Additionally, due to the anatomical variability that exists between individuals, precise placement of the toxin based solely on anatomical landmarks is still challenging. Therefore, the use of US guidance is highly recommended [39].

## 3. Anatomical Landmarks and Injection Techniques

The masseter emerges as three distinct heads from the zygomatic arch, with a tendinous structure deep inferior tendon (DIT) located profoundly to the superficial part of the muscle (Figure 1) [40].

While the superficial layer of the masseter contracts obliquely, the middle and the deeper portions are more vertical in direction and contraction [41], and the DIT is considered to block toxin diffusion from the deep portion to the superficial one, inducing overcompensation of superficial fibers and paradoxical bulging [14]. Therefore, when injecting the masseter, typically, three injection points are required, with each targeting a specific head of the masseter muscle, and injections should be evenly distributed to ensure efficacy and safety [40]. Furthermore, it is crucial to corroborate the skin surface landmarks of the masseter with the specific locations of nearby anatomical structures of interest. These structures include the facial artery and vein, the parotid gland, and the mandibular branch of the facial nerve [38]. The facial artery and vein course near the lower border of the mandible, crossing the masseter muscle superficially. The facial artery typically runs anterior to the masseter, providing blood supply to the face, while the facial vein generally follows a similar but slightly more posterior path [42]. The parotid gland, the largest salivary gland, is located posteriorly and superiorly to the masseter muscle. The gland extends over the mandible’s ramus and covers part of the masseter. The mandibular branch of the facial nerve, a motor nerve, runs inferior to the parotid gland and passes superficially to the masseter, innervating the muscles of facial expression [14]. These relationships are critical considerations during procedures involving the masseter, such as BoNT-A injections, to avoid complications and ensure precise targeting.

Having in mind the anatomic patterns of distribution of these elements, a safety zone for injection can be outlined between four lines: first line—the line from the tragus to the labial commissure; second line—the line along the jaw; third and fourth lines—the lines along the anterior and posterior edge of the masseter [41]. The three injection points, each targeting a specific head of the masseter muscle, should be evenly distributed within the lower half of the delineated quadrant (Figure 2).

## 4. Ultrasound as a Guidance Method to Enhance and Avoid Possible Complications

Ultrasonography has become widely accepted as an essential instrument in skin imaging. Recent developments have expanded its use as a method to assist BoNT-A placement in certain areas of the face [43].

While some of the facial muscles are hardly identifiable in US due to their small dimension and intermingled insertion sites, the masseter is easier to approach [44,45]. Its depiction is that of a mass with moderate homogenous echogenicity lying near the mandible’s band [46]. US imaging allows for the precise identification of all three heads of the masseter muscle. This facilitates the equitable distribution of botulinum toxin across each of these three muscle bellies (Figure 3).

While anatomical studies describe the presence of the DIT within the muscle, it is almost impossible to establish its internal location through clinical assessment [40,47]. Therefore, echography is the ideal tool to depict its precise position and architecture [19]. The DIT is visible as a hyperechoic structure that separates the superficial part of the masseter into two layers (superficial and deep) [48]. The overall architecture of the tendon can be observed in transverse US mode, whereas longitudinal-axis US images are the ones that provide the whole depiction of the tendon from origin to insertion [19,48].

Furthermore, through US examination, not only the exact location of the needle tip within the muscle can be assessed, but also the needle’s size can be adjusted based on the thickness of the subcutaneous tissue [39]. As a result, no erroneous toxin placement is performed and there is a better use of the BoNT-A units with minimum loss. Additionally, since US is the ideal instrument to canvass all areas of the face in real-time, it provides more accuracy while injecting. The changes in masseter thickness are greater after guided placement of the toxin, with subsequent better overall effect in facial slimming since it facilitates the distribution of the toxin in all regions and multilayers of the masseter [49].

US serves not only as a visualization tool but also as a quantitative measurement instrument in evaluating the outcomes of BoNT-A injections in the masseter by measuring the vertical height of the masseter muscle both before and after the injection of BoNT-A [49]. This enables the assessment of the treatment’s effectiveness by determining any reduction in the height of the masseter (Figure 4).

## 5. Unintended Possible Consequences of BoNT-A Injections in Masseter Muscle Treatment

It is crucial to be aware of potential adverse effects that may arise from incorrect injection points or depth. The most common complications include paradoxical bulging, inadvertent injection into the risorius muscle, accidental infiltration of the parotid gland, and facial paralysis [37,50]. These issues are examined in detail in the following section.

### 5.1. Paradoxical Bulging of the Masseter

Paradoxical bulging of the masseter refers to an unexpected enlargement or hypertrophy of the masseter muscle. Clinically, it presents as a visible or palpable increase in the size of the masseter muscle, leading to a bulging appearance in the lower face [50]. It causes asymmetry discomfort and may cause even functional impairment depending on the severity (Figure 5).

When administering BoNT-A injections to the masseters for medical or aesthetic purposes, it is crucial to ensure an even distribution of the toxin across all three heads of the muscle. Failure to do so can result in incomplete treatment and potential complications.

On the one hand, injecting the masseter at a single point too superficially or with insufficient dosage may only affect the superficial head of the muscle, leaving the intermediate and deep heads unaffected. Consequently, when the patient contracts the masseter, particularly during activities such as eating, the untreated heads can contract, causing the superficial head to herniate upwards. This can lead to a palpably soft masseter upon examination [40].

Conversely, injections of BoNT-A solely into the deep portion of the masseter can also present with the same complication. The presence of the DIT may obstruct the diffusion of the toxin from the deep to the superficial parts of the masseter muscle. As a result, the superficial fibers may overcompensate, leading to paradoxical bulging [14].

Utilizing US guidance can aid in identifying all three heads of the masseter, enabling physicians to accurately distribute the BoNT-A evenly [49]. By injecting each head of the masseter, the risk of inadvertent paradoxical bulging can be minimized.

BoNT-A injections into the masseter can also be beneficial in treating paradoxical bulging resulting from other various conditions, such as temporomandibular joint dysfunction, arthritis, muscle spasms, and neurological disorders affecting muscle control [50]. Additionally, congenital or acquired structural abnormalities in the jaw or surrounding tissues may contribute to paradoxical bulging, making BonT-A injections a viable treatment option in such cases [51,52].

### 5.2. Inadvertent Injection into the Risorius Muscle

The medial compartment of the masseter (the anterior third part) is commonly covered by the origin of the risorius muscle, which is a thin and variable facial muscle that originates from the fascia overlying the masseter and inserts into the skin at the angle of the mouth. It is involved in facial expressions, particularly in retracting the angle of the mouth to produce a smile [53]. On US imaging, the risorius appears as a superficial narrow band of muscle fibers running parallel to the skin, lying over the anterior portion of the masseter [54]. It may not always be distinctly visible in US due to its thin structure and variability in presence and size among individuals (Figure 6).

In their detailed anatomical study, Bae et al. examined the interactions between the risorius and masseter muscles with respect to BoNT-A injections aimed at treating masseteric hypertrophy [55]. The research involved dissecting 48 hemifaces to map the origin, insertion, and coverage of the risorius muscle over the masseter. It was discovered that the risorius muscle partially overlaid the masseter muscle in 97.8% of cases, with the coverage classified into four distinct types: type A, where the risorius covers Area III, representing the upper posterior part of the masseter (17.8% of cases); type B, covering Area VI, indicating the lower anterior part (20.0%); type C, which spans both Areas III and VI, showing extensive overlap affecting both upper and lower sections of the masseter (53.3%); and type D, which covers Areas II, III, and VI, the most comprehensive coverage impacting the upper posterior to the lower anterior parts of the masseter (6.7%) [55].

These detailed mappings underscore the importance of targeted injection practices to avoid unintended paralysis of the risorius along with further changes in facial dynamics, such as an iatrogenic asymmetric smile or grimace constraint (Figure 7) [37].

### 5.3. Accidental Infiltration of the Parotid Gland

The posterior part of the masseter is in close contact with the parotid gland, the largest salivary gland, which increases the likelihood of salivary tissue-induced trauma or inflammation (Figure 8) [38].

Furthermore, the majority of the branches of the masseteric nerve are confined to the posterior part of the masseter muscle, making this area a common target for injections [56]. Consequently, this target area is in close proximity to the parotid gland, increasing the likelihood of inadvertently affecting the gland during the procedure.

Accidental injection of BoNT-A into the parotid gland can result in several complications. One significant issue is xerostomia or dry mouth, which arises because BoNT-A inhibits acetylcholine release, thereby reducing salivary secretion [57,58]. Furthermore, inadvertent injection may lead to glandular swelling, resulting in inflammation and discomfort or pain [59]. There is also an increased risk of secondary infection if the parotid gland is inadvertently injured during the injection process [59,60].

### 5.4. Facial Paralysis

Another notable complication is facial paralysis, which can particularly affect the lower face. The facial nerve (cranial nerve VII) is responsible for innervating the muscles of facial expression [61]. It emerges from the stylomastoid foramen and enters the parotid gland, where it divides into five major branches: temporal, zygomatic, buccal, mandibular, and cervical. These branches radiate through the parotid gland and extend to various parts of the face to control muscle movements involved in expressions such as smiling, frowning, and blinking [61].

Due to this close anatomical relationship, these nerve branches are at risk of inadvertent injury during procedures like botulinum toxin injections into the masseter muscle. If the injection is misplaced or diffuses into the parotid gland, it can affect the facial nerve branches, leading to temporary facial paralysis or weakness. This is caused by compression or mild trauma to the nerve and corresponds to a type of peripheral nerve injury called neuropraxia, which is characterized by a temporary block in nerve conduction without structural damage to the nerve fibers themselves. Since the axon remains intact, recovery is usually complete once the myelin sheath repairs itself, typically within a few weeks to months. While rare, such complications underline the importance of precise anatomical knowledge and careful technique when performing BoNT-A injections [13].

Taking all these aspects into account, a US-based evaluation is crucial to identify the structures at risk and avoid complications. The real-time visualization of anatomical variants and boundaries of the muscle conveys a higher safety and reliability than the conventional method.

## 6. Potential Limitations and Considerations

While the use of ultrasound guidance in botulinum toxin injections offers significant advantages in terms of precision and safety, there are important limitations and considerations that should be acknowledged to provide a balanced perspective.

Cost and Accessibility of Ultrasound: One of the primary limitations of implementing ultrasound guidance in botulinum toxin injections is the cost associated with acquiring and maintaining the equipment. High-quality ultrasound devices can be expensive, with prices varying depending on the model and features required for facial imaging. Additionally, the availability of ultrasound equipment is not uniform across different regions, with significant disparities in access, particularly in developing countries. These financial and logistical barriers may prevent widespread adoption of ultrasound-guided techniques, potentially limiting access for both practitioners and patients who could benefit from this technology [18];The Learning Curve: Effectively utilizing ultrasound guidance for botulinum toxin injections is not without its challenges, particularly regarding the learning curve. Proficiency in ultrasound imaging requires specific training and experience, as well as a deep understanding of facial anatomy and the nuances of sonographic interpretation. Practitioners must invest time and resources into acquiring these skills to use ultrasound effectively during injections. The importance of proper training cannot be overstated, as improper use of ultrasound may lead to inaccurate injections, potentially increasing the risk of adverse effects [19];Sterility Issues: An important but often overlooked consideration in the use of ultrasound for botulinum toxin injections is the issue of sterility. For the procedure to remain sterile during simultaneous ultrasound imaging and injection, the practitioner must use a sterile ultrasound gel. However, this practice is not universally adopted, and there is limited discussion in the literature regarding the potential risks associated with non-sterile gel use [41].

## 7. Long-Term Effects of Repeated BoNT-A Injections

The long-term effects of repeated BoNT-A injections into the masseter muscle represent an area of ongoing research, particularly in light of the increasing application of these treatments for both medical and aesthetic purposes. While BoNT-A has demonstrated significant efficacy in reducing muscle hypertrophy and reshaping the jawline, there are important considerations regarding its impact on muscle and bone health over extended periods.

One of the primary benefits of repeated BoNT-A injections is the induction of muscular plasticity, which refers to the ability of muscle tissue to adapt structurally and functionally in response to these treatments. Over time, BoNT-A leads to a reduction in muscle activity, leading to reduced muscle bulk and alterations in muscle fiber composition, which contribute to a more balanced facial contour and muscle function. This effect is particularly beneficial in patients with conditions like masseter hypertrophy and bruxism, where excessive muscle activity poses functional and aesthetic challenges [62,63,64,65].

However, alongside these benefits, there are concerns about the potential adverse effects of BoNT-A on bone health, particularly the phenomenon of bone resorption at the mandibular angle. Repeated injections may weaken the masseter muscle to the extent that it affects the underlying bone structure, potentially leading to changes in bone density and morphology. A study raised concerns that BoNT-A could accelerate osteopenia at the mandibular angle, especially in individuals receiving long-term treatment for cosmetic purposes [63]. The relationship between muscle activity and bone homeostasis is well-documented, with muscle contraction being necessary for maintaining bone integrity through mechano-transduction processes [66]. The risk of bone resorption appears to be more significant in certain populations, particularly post-menopausal females, where reduced bone density is already a concern [67].

## 8. Conclusions

BoNT-A injections in the masseter muscle demonstrate considerable versatility and efficacy in both medical and aesthetic applications, offering significant benefits such as muscle relaxation and contouring enhancements. The use of US guidance is highly recommended as it enhances the efficiency of these injections, ensuring more effective distribution of the toxin and resulting in improved therapeutic outcomes. By enabling precise injection placement, US guidance minimizes potential adverse effects, including asymmetry, paradoxical bulging, and accidental involvement of nearby structures such as the parotid gland or facial nerve.

As the application of BoNT-A continues to expand both in medical and aesthetic fields, it is imperative to focus on optimizing therapeutic outcomes while mitigating risks. This includes refining injection techniques and deepening our understanding of the long-term effects of these treatments. Ongoing research and collaboration will be essential to advance these practices and enhance patient care.

## Figures and Tables

**Figure 1 toxins-16-00413-f001:**
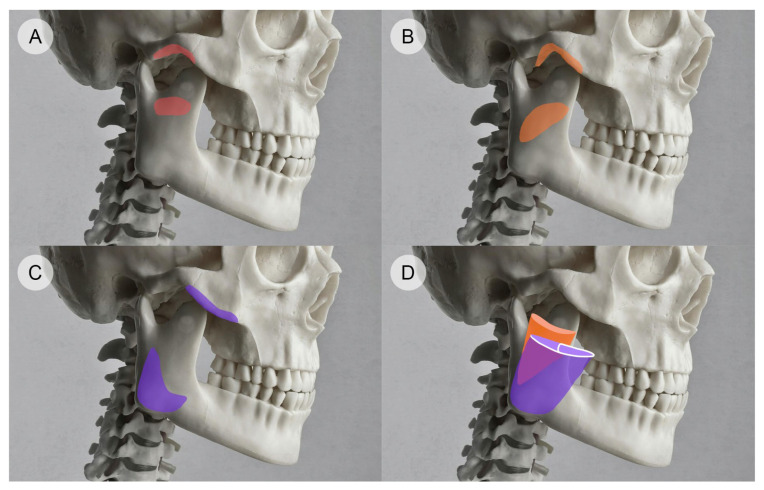
Schematic illustration depicting the three heads of the masseter muscle originating from the zygomatic arch: the deep head, origin, and insertion (**A**), the intermediate head, origin, and insertion (**B**), and the superficial head, origin, and insertion (**C**). The deep inferior tendon (highlighted in white) separates the deep portion from the superficial one (**D**).

**Figure 2 toxins-16-00413-f002:**
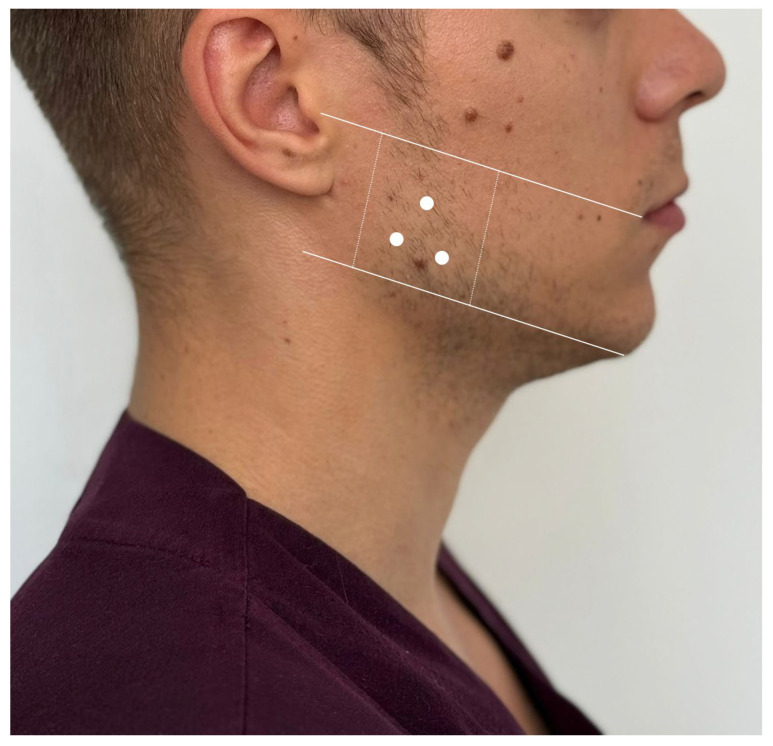
Anatomically guided safe zone for BoNT-A injections in the masseter muscle: schematic representation with three targeted injection points.

**Figure 3 toxins-16-00413-f003:**
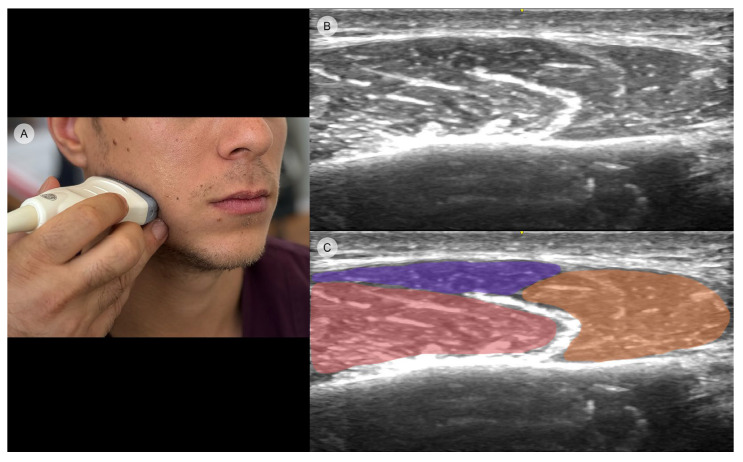
Ultrasound imaging of the masseter muscle demonstrating its three distinct heads (GE Venue Ultrasound, 12L-RS transducer, 3.4–12.6 MHz): (**A**) clinical image illustrating transducer placement in the transverse position; (**B**) original ultrasound image showing the muscle structure; (**C**) edited ultrasound image with color highlights distinguishing the deep head (orange), intermediate head (red), and superficial head (purple) for clearer visualization.

**Figure 4 toxins-16-00413-f004:**
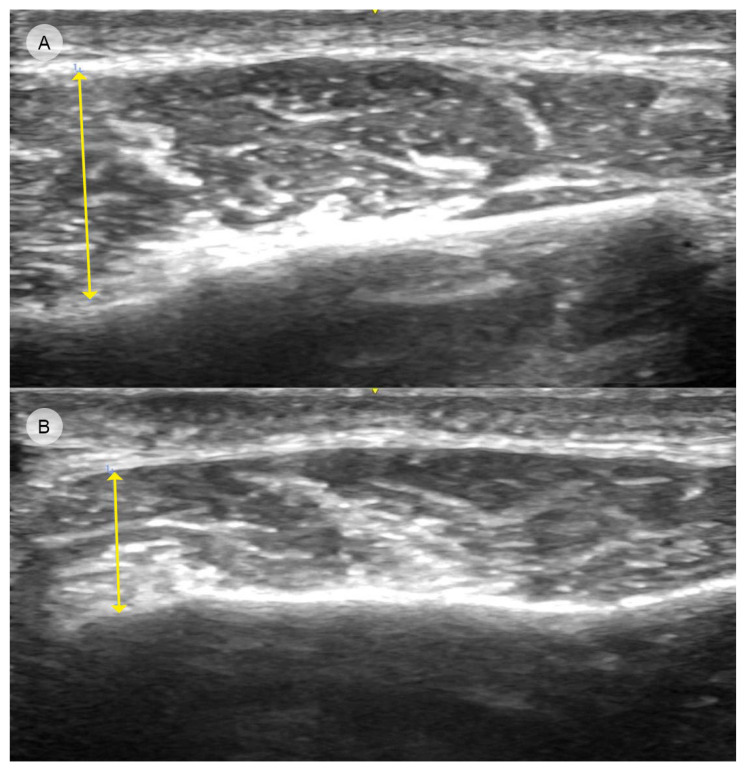
Ultrasound images showing the reduction in masseter muscle thickness before (**A**) and three weeks after (**B**) BoNT-A injection (GE Venue Ultrasound, 12L-RS transducer, 3.4–12.6 MHz). The arrows indicate the maximum vertical height of the muscle in both images.

**Figure 5 toxins-16-00413-f005:**
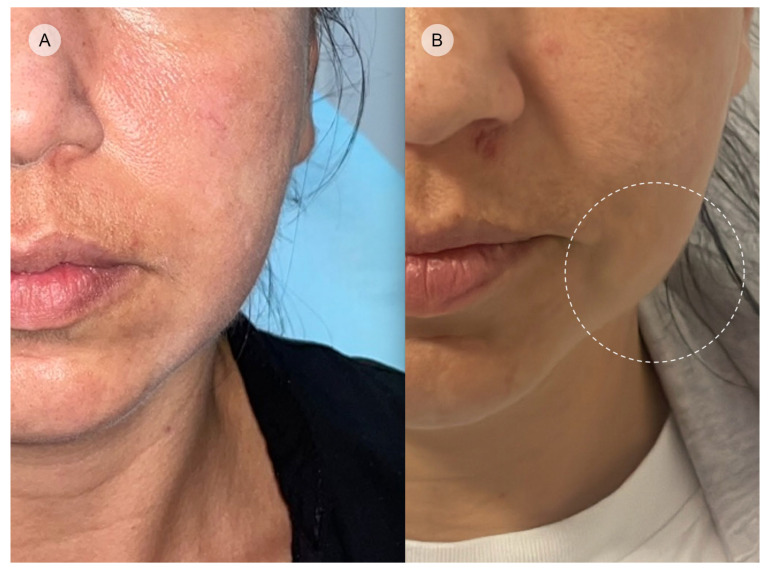
Paradoxical bulging of the masseter muscle: (**A**) pre-treatment clinical presentation of the lower face; (**B**) post-treatment image showing visible and palpable asymmetric masseter bulging (circled).

**Figure 6 toxins-16-00413-f006:**
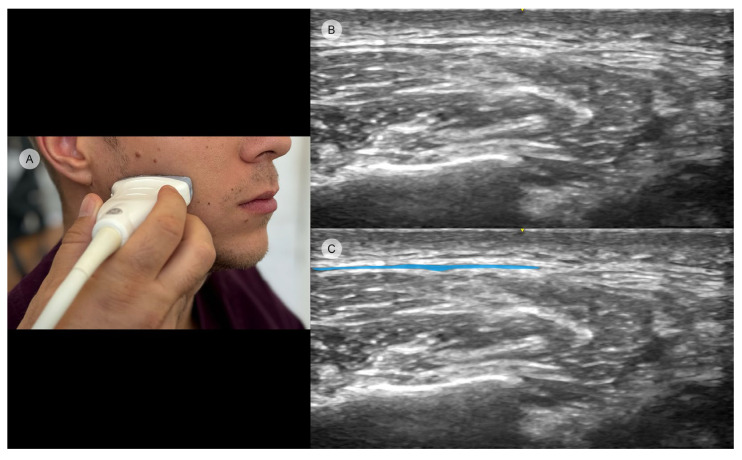
Ultrasound image depicting the risorius muscle anterior to the medial compartment of the masseter (GE Venue Ultrasound, 12L-RS transducer, 3.4–12.6 MHz): (**A**) clinical image illustrating the transducer placement in the transverse position; (**B**) original ultrasound image showing both the risorius and masseter muscles; (**C**) edited ultrasound image with the risorius muscle highlighted in blue for clearer identification.

**Figure 7 toxins-16-00413-f007:**
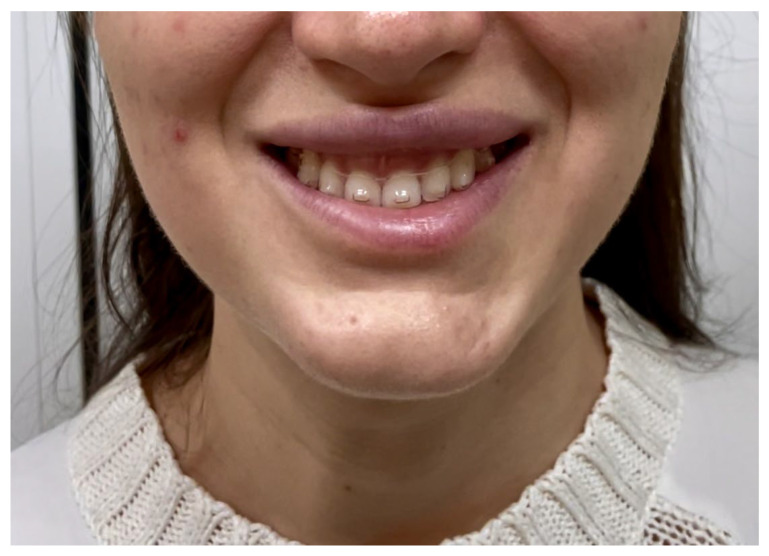
Clinical manifestation of risorius muscle involvement after BoNT-A injection into the masseter muscle. The figure shows a patient attempting a full smile, illustrating asymmetry due to the unintended weakening of the risorius muscle on one side. Additionally, there is a noticeable misalignment where the center of the gingiva does not correspond with the center of the upper lip, highlighting the asymmetry in the lower facial region.

**Figure 8 toxins-16-00413-f008:**
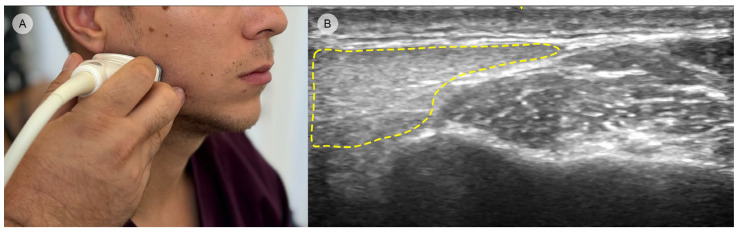
Ultrasound image showing the parotid gland in close contact with the posterior region of the masseter (GE Venue Ultrasound, 12L-RS transducer, 3.4–12.6 MHz): (**A**) clinical image showing the transducer placement in the transverse position; (**B**) edited ultrasound image with the parotid gland highlighted (yellow dashed area).

## Data Availability

This review summarizes data reported in the literature and it does not report primary data.

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
