# Peer review of "Ultrasound-Guided Botulinum Toxin-A Injections into the Masseter Muscle for Both Medical and Aesthetic Purposes"

_toxins, 2024, doi:10.3390/toxins16100413_

Round 1

Reviewer 1 Report

Comments and Suggestions for Authors

The authors of the research study entitled “Ultrasound-Guided Botulinum Toxin-A Injections for both Medical and Aesthetic Purposes” aim to provide a comprehensive review regarding the ultrasound-guided intramuscular injection fused for masseter muscle interventions in humans, for either medical or aesthetic purposes. To achieve this, the authors have written a literature review including explanations about masseter muscle anatomy and its relationships with other relevant structures, rationale to use botulinum toxin type A (BoNTA) interventions, potential consequences of BoNTA injection in the masseter muscle in humans, and suggestions for future addresses on the topic. The topic is relevant to the field, since there is a need to understand how to improve patient safety while doing invasive interventions in the masticatory system. However, there are some limitations in this study, that are listed as major concerns in the comments to the authors, as follows: 

Major concerns: 

1.     A comprehensive review should include not only a general explanation of the chapters, but an in-deep description for each category, such as:

-       Ultrasound principles and musculoskeletal applications in dentistry

-       Masseter anatomy and physiology

-       BoNTA as therapeutic tool and its mechanism of action

-       Potential adverse effects from animal to clinical studies

2.     Please consider including the term dental/dentistry in the tittle.

3.     Please describe other techniques assisted by ultrasound such as local anesthesia, dry needle technique, among others.

4.     Please include conditions of interest other than bruxism with a brief description of the physio-pathological features and how BoNTA plays a role on their management.

5.     Please consider improving the visual impact of the review by including 3D images of the main vessels and nerves surrounding the masseter muscle in humans.

6.     Also, since US-guided techniques are performed in real time, it is advisable to include a clinical video, protecting the identity of the patients, which is ethically correct for clinical photos and videos. 

Comments on the Quality of English Language

Please review the accepted nomenclature for anatomical structures and the tittle for each section.

Author Response

Comment 1: A comprehensive review should include not only a general explanation of the chapters, but an in-deep description for each category, such as:

  • Ultrasound principles and musculoskeletal applications in dentistry

Response: While we acknowledge the importance of ultrasound principles and musculoskeletal applications in dentistry, we respectfully believe that this topic lies beyond the scope of our paper. Our manuscript is written from the perspective of dermatologists and physical and rehabilitation medicine specialists, with a primary focus on muscular ultrasound-guided Botulinum Toxin-A injections for both medical and aesthetic purposes. In our country, dentists are not authorized to perform BonT-A injections, even in cases involving overlapping medical fields, such as certain dental pathologies. Therefore, the scope of our paper remains within the expertise and practice domains of our specialties, rather than incorporating dental perspectives.

-       Masseter anatomy and physiology

Response: Thank you for your comment. Our manuscript includes a comprehensive chapter dedicated to the anatomy and physiology of the masseter muscle. Please see Lines 130-167. We believe the content provided adequately addresses this topic and is sufficient within the context of our paper.

-       BoNT-A as therapeutic tool and its mechanism of action

Response: Thank you for your observation. We have already addressed the mechanism of action of BoNT-A in the introduction section. We kindly refer you to lines 40-47 for further details.

Additionally, the second chapter thoroughly discusses BoNT-A as a therapeutic tool for various pathologies involving the masseter muscle, such as masseter hypertrophy, bruxism, TMDs, oromandibular dystonia, MPS, and TTH. Each of these conditions is covered in detail, including the specific role BoNT-A plays in their treatment. Please see Lines 57-129.

-       Potential adverse effects from animal to clinical studies

Response: Thank you for your comment. However, it seems there may have been a misunderstanding. The method of muscular BoNT-A injections has been widely and safely used in humans for many years across various medical and aesthetic indications. Our manuscript is not focused on experimental animal studies, but rather on enhancing the safety of these injections through the use of ultrasound guidance to improve precision. The goal of our paper is to provide a detailed approach to increase accuracy and minimize potential adverse effects in human applications

Comment 2.     Please consider including the term dental/dentistry in the tittle.

Response : Thank you for your suggestion. However, as previously mentioned, our manuscript is not written from a dental perspective

Comment 3.     Please describe other techniques assisted by ultrasound such as local anesthesia, dry needle technique, among others.

Response : Thank you for your comment. While we acknowledge the value of techniques such as ultrasound-assisted local anesthesia and dry needling, we do not see their relevance to the scope of our paper. Our focus is specifically on enhancing the precision and safety of Bon T-A injections in the masseter through the use of ultrasound guidance, and not on other ultrasound-assisted techniques. Therefore, we believe it is best to maintain the paper's focus on its intended subject matter.

Comment 4.     Please include conditions of interest other than bruxism with a brief description of the physio-pathological features and how BoNT-A plays a role on their management.

Response: Thank you for your comment. The second chapter is specifically focused on various conditions beyond bruxism, such as masseter hypertrophy, temporomandibular disorders (TMDs), oromandibular dystonia, myofascial pain syndrome, and tension-type headaches. Please see lines 57-129.

  1. Please consider improving the visual impact of the review by including 3D images of the main vessels and nerves surrounding the masseter muscle in humans.

Response: Thank you for your suggestion. Our manuscript already includes 8 figures, consisting of graphics, clinical images, and ultrasound visuals, which we believe are sufficient to support our discussion. Creating new 3D images would require significant additional effort and graphical expertise, which we do not have the resources for at this time. However, we appreciate your feedback and will certainly consider incorporating 3D visuals in future papers.

  1. Also, since US-guided techniques are performed in real time, it is advisable to include a clinical video, protecting the identity of the patients, which is ethically correct for clinical photos and videos.

Response: Thank you for your suggestion. Producing clinical videos would require additional time and specialized IT/graphics skills, which our current team does not possess. However, we appreciate your feedback and will consider incorporating clinical videos in future papers, possibly by recruiting collaborators with video editing expertise.

Reviewer 2 Report

Comments and Suggestions for Authors

Please specify better the mechanism of nerve neuropraxia complication mentioned in the article by the authors. It's unclear whether they mean that it is due to mechanical damage to the nerve caused by the injection with needle or by the action of the botulinum toxin on the nerve. Note that neuropraxia is defined as a temporary loss of motor and sensory function due to the blockage in nerve conduction caused by compression or mild trauma of the nerve. In chapter 5.4 the authors don't explain the mechanism of the risk of facial paralysis and it seems related to the botulinum toxin diffusion to the nearby branches of the facial nerve inside the parotid gland.

More details about the characteristics of the ultrasound machine, probes and scan techniques used by the authors  are necessary.

Chapter 6 "Future directions"  doesn't seem strictly correlated with the topic of the article. 

Author Response

Comment 1: Please specify better the mechanism of nerve neuropraxia complication mentioned in the article by the authors. It's unclear whether they mean that it is due to mechanical damage to the nerve caused by the injection with needle or by the action of the botulinum toxin on the nerve. Note that neuropraxia is defined as a temporary loss of motor and sensory function due to the blockage in nerve conduction caused by compression or mild trauma of the nerve. In chapter 5.4 the authors don't explain the mechanism of the risk of facial paralysis and it seems related to the botulinum toxin diffusion to the nearby branches of the facial nerve inside the parotid gland.

Reponse: Thank you for your valuable feedback. We have addressed this issue in the revised manuscript, specifically in lines 307-316, where we explain the mechanism more clearly.

Comment 2: More details about the characteristics of the ultrasound machine, probes and scan techniques used by the authors  are necessary.

Reponse: Thank you for your suggestion. You are absolutely right, and these specifications can now be found in the description accompanying each ultrasound image in the manuscript.

Comment 3: Chapter 6 "Future directions"  doesn't seem strictly correlated with the topic of the article. 

Response: Thank you for your observation. We understand your concern, and we have revised Chapter 6 accordingly. The content has been adapted and transformed into Chapter 7, titled 'Long-Term Effects of Repeated BoNT-A Injections,' which we believe is now more closely aligned with the topic of the article and provides a clearer connection to the main discussion.

Reviewer 3 Report

Comments and Suggestions for Authors

The manuscript "Ultrasound-Guided Botulinum Toxin-A Injections for both Medical and Aesthetic Purposes" explores the applications of ultrasound-guided Botulinum toxin type A injections for treating masseter muscle conditions.

The authors claim that using ultrasound guidance significantly improves the accuracy and safety of these injections compared to traditional methods. The manuscript describes the anatomy of the masseter muscle, highlighting anatomical landmarks crucial for successful injections.

Furthermore, it discusses these injections' medical and aesthetic benefits, such as treating masseter hypertrophy, bruxism, temporomandibular disorders, and facial slimming. Finally, the authors emphasize the importance of ultrasound guidance in minimizing potential complications associated with botulinum toxin-A injections in the masseter muscle.

Comments

1. The manuscript exclusively discusses the treatment of masseter hypertrophy. Therefore, the title should be revised to reflect the manuscript's content accurately.

2. Please provide the details of the ultrasound settings, including the frequency, type of probe, and the device's product name shown in the figures.

3. While the manuscript provides a comprehensive overview of ultrasound-guided Botulinum toxin-A injections for the masseter muscle, one weakness is that it primarily focuses on the advantages of this technique. The manuscript could benefit from a more thorough discussion of this procedure's limitations and potential drawbacks to present a more balanced perspective. This could include:

- Cost and accessibility of ultrasound guidance: Ultrasound equipment can be expensive, and its availability may vary, potentially limiting access to this technique for practitioners and patients, especially in developing countries.

- The learning curve for practitioners: Effectively using ultrasound guidance requires specific training and experience. The manuscript could benefit from acknowledging this learning curve and emphasizing the importance of proper practitioner training. The authors could recommend a review article or textbook that would help beginners learn to use ultrasound imaging on the face and neck.

- Sterility issues: Injecting during imaging is only sterile if the practitioner uses a sterile sono gel. This issue has not been described in the literature, and many clinicians hesitate to perform simultaneous ultrasound imaging during cosmetic botulinum toxin injections. 

By acknowledging these limitations, the manuscript would provide a more comprehensive and balanced view of ultrasound-guided botulinum toxin injections for the masseter muscle.

Author Response

Comment 1: The manuscript exclusively discusses the treatment of masseter hypertrophy. Therefore, the title should be revised to reflect the manuscript's content accurately.

Response: Thank you for your feedback. We agree with your suggestion and have modified the title of the manuscript to: 'Ultrasound-Guided Botulinum Toxin-A Injections into the Masseter Muscle for both Medical and Aesthetic Purposes,' which we feel better reflects the content of the paper.

Comment 2: Please provide the details of the ultrasound settings, including the frequency, type of probe, and the device's product name shown in the figures.

Reponse: Thank you for your suggestion. You are absolutely right, and these specifications can now be found in the description accompanying each ultrasound image in the manuscript.

Comment 3. While the manuscript provides a comprehensive overview of ultrasound-guided Botulinum toxin-A injections for the masseter muscle, one weakness is that it primarily focuses on the advantages of this technique. The manuscript could benefit from a more thorough discussion of this procedure's limitations and potential drawbacks to present a more balanced perspective. This could include:

- Cost and accessibility of ultrasound guidance: Ultrasound equipment can be expensive, and its availability may vary, potentially limiting access to this technique for practitioners and patients, especially in developing countries.

- The learning curve for practitioners: Effectively using ultrasound guidance requires specific training and experience. The manuscript could benefit from acknowledging this learning curve and emphasizing the importance of proper practitioner training. The authors could recommend a review article or textbook that would help beginners learn to use ultrasound imaging on the face and neck.

- Sterility issues: Injecting during imaging is only sterile if the practitioner uses a sterile sono gel. This issue has not been described in the literature, and many clinicians hesitate to perform simultaneous ultrasound imaging during cosmetic botulinum toxin injections. 

Response: We agree that discussing the limitations and potential drawbacks of ultrasound-guided Botulinum Toxin-A injections is essential for presenting a more balanced perspective. In response, we have created a new chapter, entitled Chapter 6. Potential Limitations and Considerations”. Please refer to lines 321-348 for the detailed discussion.